# REVIEW ARTICLE: THE USE OF REMOTELY PILOTED AIRCRAFT SYSTEMS (RPAS) FOR NATURAL HAZARDS MONITORING AND MANAGEMENT

Daniele Giordan[1], Yuichi Hayakawa[2], Francesco Nex[3], Fabio Remondino[4], Paolo Tarolli[5]

[1]Istituto di Ricerca per la Protezione Idrogeologica, Consiglio Nazionale delle Ricerche, Italy
[2]Center for Spatial Information Science, The University of Tokyo, Japan
[3]University of Twente, Faculty of Geo-Information Science and Earth Observation (ITC), The Netherlands
[4]3D Optical Metrology (3DOM) Unit, Bruno Kessler Foundation (FBK), Trento, Italy
[5]Department of Land, Environment, Agriculture and Forestry, University of Padova, Italy

## ABSTRACT

The number of scientific studies that consider possible applications of Remotely Piloted Aircraft Systems (RPAS) for the management of natural hazards effects and the identification of occurred damages are strongly increased in last decade. Nowadays, in the scientific community, the use of these systems is not a novelty, but a deeper analysis of literature shows a lack of codified complex methodologies that can be used not only for scientific experiments but also for normal codified emergency operations. RPAS can acquire on-demand ultra-high resolution images that can be used for the identification of active processes like landslides or volcanic activities but also for the definition of effects of earthquakes, wildfires and floods. In this paper, we present a review of published literature that describes experimental methodologies developed for the study and monitoring of natural hazards.

## 1. INTRODUCTION

In last three decades, the number of natural disasters showed a positive trend with an increase in the number of affected populations. Disasters not only affected the poor and characteristically more vulnerable countries but also those thought to be better protected. Annual Disaster Statistical Review describes recent impacts of natural disasters over population and reports 342 natural triggered disasters in 2016 (Guha-Sapir et al., 2017). This is less than the annual average disaster frequency observed from 2006 to 2015 (376.4 events), however natural disasters is still responsible for a high number of casualties (8,733 death). In the period 2006-2015, the average number of casualities annaly caused by natural disasters is 69,827. In 2016, hydrological disasters (177) had the largest share in natural disaster occurrence (51.8%), followed by meteorological disasters (96; 28.1%), climatological disasters (38; 11.1%) and geophysical disasters (31; 9.1%) (Guha-Sapir et al., 2017). To face these disasters, one of the most important solutions is the use of systems able to provide an adequate level of information for correctly understanding these events and their evolution. In this context, survey and monitoring of natural hazards gained in importance. In particular, during the emergency phase it is very important to evaluate and control the phenomenon evolution, preferably operating in near real time or real time, and consequently, use this information for a better risk scenario assessment. The available acquired data must be processed rapidly to ensure the emergency services and decision makers promptly.

Recently, the use of remote sensing (satellite and airborne platform) in the field of natural hazards and disasters has become common, also supported by the increase in geospatial technologies and the ability to provide and process up-to-date imagery (Joyce et al., 2009; Tarolli, 2014). Remotely sensed data play an integral role in predicting hazard events such as floods and landslides, subsidence events and other ground instabilities. Because their acquisition mode and capability for repetitive observations, the data acquired at different dates and high spatial resolution can be considered as an effective complementary tool for field techniques to derive information on landscape evolution and activity over wide areas.

In the contest of remote sensing research, recent technological developments have increased in the field of Remotely Piloted Aircraft Systems (RPAS) becoming more common and widespread in civil and commercial context (Bendea et al., 2008). In particular, the development of photogrammetry and technologies associated (i.e. integrated camera systems like compact cameras, industrial grade cameras, video cameras, single-lens reflex (SLR) digital cameras and GNSS/INS systems) allow to use of RPAS platforms in various applications as alternative to the traditional remote sensing method for topographic mapping or detailed 3D recording of ground information and a valid complementary solution to terrestrial acquisitions too (Nex and Remondino, 2014) (Fig.1).

RPAS systems present some advantages in comparison to traditional platforms and, in particular, they could be competitive thanks to their versatility in the flight execution (Gomez and Purdie, 2016). Mini/micro RPAS are the most diffused for civil purposes, and they can fly at low altitudes according to limitations defined by national aviation security agencies and be easy transported into the disaster area. Foldable Systems fits easily into a daypack and can be transported safely as hand luggage. This advantage is particularly important for first responder teams like UNDAC or similar. Stöcker et al. (2017) published a review of different state regulations that are characterized by several differences regarding requirements, distance from the takeoff and maximum altitude. Another important added value of RPAS is their adaptability that allows their use in various typologies of missions, and in particular for monitoring operations in remote and dangerous areas (Obanawa et al., 2014). The possibility to carry out flight operations at lower costs compared to ones required by traditional aircraft is also a fundamental advantage. Limited operating costs make these systems also convenient for multi-temporal applications where it is often necessary to acquire information on an active process (like a landslide) over the time.  A comparison between the use of satellite images, traditional aircraft and RPAS has been presented and discussed by Fiorucci et al. (2018) for landslides applications and by Giordan et al., (2017) for the identification of flooded areas. These comparisons show that RPAS are a good solution for the on demand acquisition of high resolution images over limited areas.

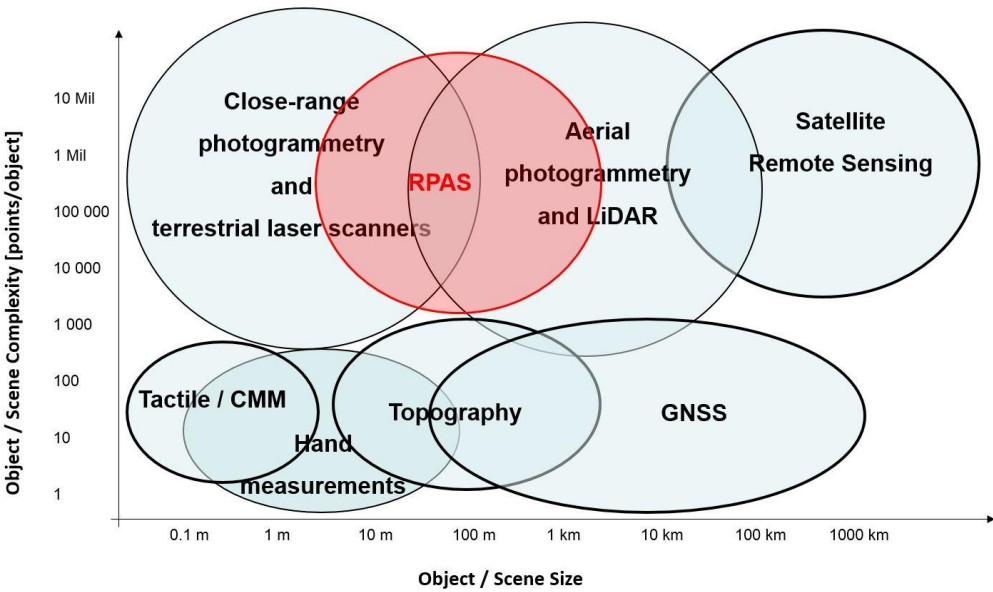

Figure 1. Available geomatics techniques, sensors, and platforms for topographic mapping or detailed 3D recording of ground information, according to the scene dimensions and complexity (modified from Nex and Remondino, 2014).

RPASs are used in several fields as agriculture, forestry, archaeology and architecture, traffic monitoring, environment and emergency management. In particular, in the field of emergency assistance and management, RPAS platforms are used to reliably and fast collect data of inaccessible areas (Huang et al., 2017). Collected data can be mostly images but also gas concentrations or radioactivity levels as demonstrated by the tragic event in Fukushima (Sanada and Torii, 2015; Martin et al., 2016). Focusing on image collection, they can be used for early impact assessment, to inspect collapsed buildings and to evaluate structural damages on common infrastructures (Chou et al. 2010; Molina et al. 2012; Murphy et al., 2008; Pratt et al., 2009) or cultural heritage sites (Pollefeys et al., 2001; Manfredini et al., 2012; Koutsoudisa et al., 2014; Lazzari et al., 2017). Environmental and geological monitoring can profit from fast multi-temporal acquisitions delivering high-resolution images (Thamm and Judex 2006; Niethammer et al. 2010). RPAS can be considered a good solution also for mapping and monitoring different active processes at the earth surface (Fonstad et al., 2013; Piras et al., 2017; Feurer et al., 2017; Hayakawa et al., 2018) such as: glaciers (Immerzel et al., 2014, Ryan et al., 2015; Fugazza et al., 2017), Antarctic moss beds (Lucieer et al., 2014b), costal areas (Delacourt et al., 2009; Klemas, 2015), Interseismic deformations (Deffontaines et al., 2017; 2018), river morphodynamic (Gomez and Purdie, 2016; Jaud et al., 2016; Aicardi et al., 2017; Bolognesi et al., 2016; Benassai et al., 2017), debri flows (Wen et al., 2011), and river channel vegetation (Dunford et al., 2009).

The incredible diffusion of RPAS has pushed many companies to develop dedicated sensors for these platforms. Besides the conventional RGB cameras other camera sensors are nowadays available on the market. Multi- and hyper-spectral cameras, as well as thermal sensors, have been miniaturized and customized to be hosted on many platforms.

The general workflow of a UAV acquisition is presented in Figure 2 below. The resolution of the images, the extension of the area as well as the goal of the flight are the main constraints that affect the selection of the platform and the typology of the sensor. Large areas can be flown using fixed wing (or hybrid) solutions able to acquire nadir images in a fast and efficient way. Small areas or complex objects (like steep slopes or buildings) should be acquired using rotor RPAS as they are usually slower but they allow the acquisition of oblique views. If the information different from the visible band is needed, the RPAS can host one or more

sensors acquiring in different bands. The flight mission can be planned using dedicated software: they range from simple apps installed on smartphones in the low-cost solutions, to laptops connected to directional antennas and remote controls for the most sophisticated platforms. According to the typology of the platform, different GNSS and IMU can be installed. Low-cost solutions are usually able to give positions with few meters accuracy and need GCP (Ground Control Points) to geo-reference the images. On the other hand, most expensive solutions install double frequency GNSS receivers with the possibility to get accurate geo-referencing thanks to Real Time Kinematic (RTK) or Post Processing Kinematic (PPK) corrections. The use of GCP and different GNSS solutions is a fundamental point. Gercke and Przybilla (2016) presented the effect of RTK-GNSS and cross flight patterns, and Nocerino et al., (2013) presented an evaluation about RPAS processing results quality considering: i) the use of GCPs, ii) different photogrammetric procedures, iii) different network configurations. If a quick mapping is needed, the information delivered by the navigation system can be directly used to stitch the images and produce a rough image mosaicking (Chang-chun et al., 2011). In the alternative, the typical photogrammetric process is followed: (i) image orientation, (ii) DSM generation and (iii) orthophoto generation. The position (geo-referencing) and the attitude (rotation towards the coordinates system) of each acquisition is obtained by estimating the image orientation. In the dense point cloud generation, 3D point clouds are generated from a set of images, while the orthophoto is generated in the last step combining the oriented images projected on the generated point cloud, leading to orthorectified images (Turner et al., 2012). Point clouds can be very often converted in Digital Surface Models (DSM), and Digital Terrain Models (DTM) can be extracted removing the off ground regions (mainly buildings and trees). In real applications, many parameters can influenced the final resolution of DSM/DTM and ortophoto like: real GSD (Nocerino et al., 2013) interior and exterior orientation parameters (Kraft et al., 2016), overlap of images, flight strip configuration and used SfM-Software (Nex et al., 2015).

 In particular during emergencies, the time required for the image dataset processing can be a critical point. For this reason, the development of fast mosaicking methods as MACS, for a real time mapping applications (Lehmann et al., 2011), or VABENE++, developed by German Aerospace Center for real time traffic management (Detzer et al., 2015).

The outputs from the last two steps (point clouds and true-orthophotos) as well as the original images are very often used as input in the scene understanding process: classification of the scene or extraction of features (i.e. objects) of interest using machine learning techniques are the most common applications. 3D models can also be generated using the point cloud and the oriented images to texturize the model.

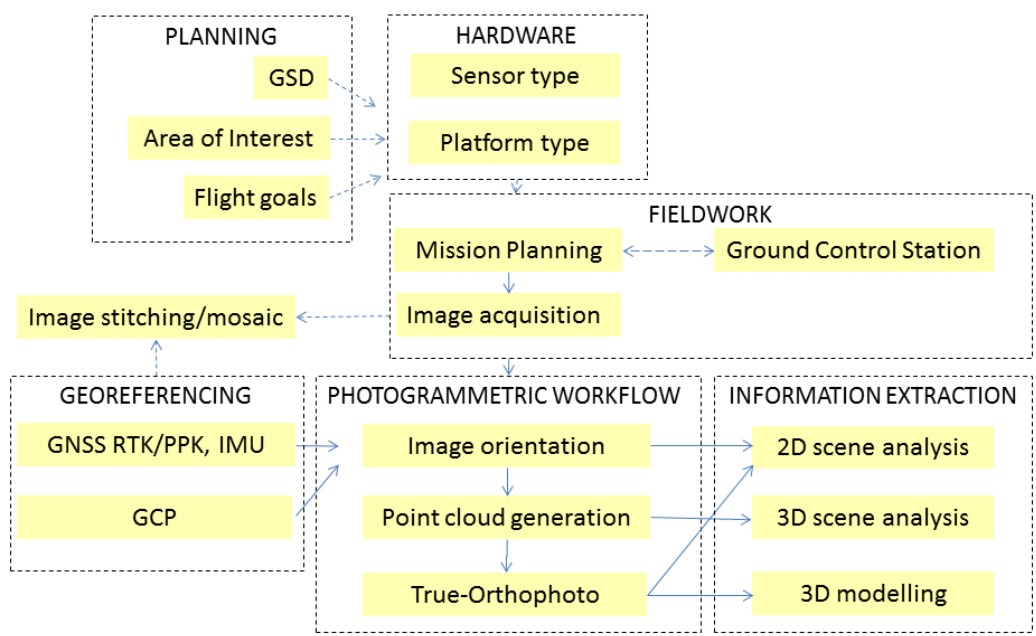

Figure 2. Acquisition and processing of RPAS images: general workflow.

In this paper, the authors present an analysis and evaluation concerning the use of RPAS as alternative monitoring technique to the traditional methods, relating to the natural hazard scenarios. The main goal is to define and test the feasibility of a set of methodologies that can be used in the monitoring and mapping activities. The study is focused in particular on the use of mini and micro RPAS systems (Table 1). The following table listed the technical specifications of these two RPAS categories, again based on the current classification by UVS (Unmanned Vehicle Systems) International. Most of the mini or micro RPAS systems available integrate a flight control system, which autonomously stabilizes these platforms and enables the remotely controlled navigation. Additionally, they can integrate an autopilot, which allows an autonomous flight based on predefined waypoints. For the monitoring and mapping applications, mini- or micro RPAS systems are very useful as cost-efficient platforms for capturing real-time close-range imagery. These platforms can reach the area of investigation and take several photos and videos from several points and different angles of view (Gomez and Kato, 2014). For mapping applications, it is also possible to use this flight control data to geo-register the captured payload sensor data like still images or video streams (Eugster and Nebiker, 2008).

Table 1. Classification of mini and micro UAV systems, according to UVS International (UVS International, 2018)

| Category | Max. Take Of Weight | Max. Flight Altitude | Endurance | Data Link Range |
|---|---|---|---|---|
| Mini | <30kg | 150-300m | <2h | <10km |
| Micro | <5Kg | 250m | 1h | <10km |

## 2. USE OF RPAS FOR NATURAL HAZARDS DETECTION AND MONITORING

Gomez and Purdie (2016) published a detailed analysis of the use of RPAS for hazards and disaster risk monitoring. In our paper, we focused our attention on the most dangerous natural hazards that can be

analyzed using RPAS. According to the definitions used by Annual Disaster Statistical Review (Guha-Sapir et al., 2017), the paper considers in particular:  i) landslides, ii) floods iii) earthquakes v) volcanic activity vi) wildfires. For each considered category of natural hazard, the paper presents a review of a large list of published papers (171 papers), analyzing proposed methodologies and provided results, and underlining strengths and limitations in the use of RPAS. The aims of this paper is the description of possible use of RPAS in considered natural hazards, describing a general methodology for the use of these systems in different contexts merging all previous published experiences.

## 2.1 Landslides

Landslides are one of the major natural hazards that produce each year enormous property damage regarding both direct and indirect costs. Landslides are rock, earth or debris flows on slopes due to gravity. The event can be triggered by a variety of external elements, such as intense rainfall, water level change, storm waves or rapid stream erosion that cause a rapid increase in shear stress or decrease in shear strength of slope-forming materials. Moreover, the pressures of increasing population and urbanization, human activities such as deforestation or excavation of slopes for road cuts and building sites, etc., have become important triggers for landslide occurrence. Because the factors affecting landslides can be geophysical or human-made, they can occur in developed and undeveloped areas.

In the field of natural hazards, the use of RPAS for landslides study and monitoring represents one of the most common applications. The number of papers that present case studies or possible methodologies dedicated to this topic has strongly increased in last few years and now the available bibliography offers a good representation of possible approaches and technical solutions.

When a landslide occurs, the first information to be provided is the extent of the area affected by the event (figure 3). The landslide impact extent is usually done based on detailed optical images acquired after the event. From these acquisitions, it is possible to derive Digital Elevation Models (DEMs) and orthophotos that allow detecting main changes in geomorphological figures (Fan et al., 2017; Chang et al., 2017). In this scenario, the use of the mini-micro RPAS is practical for small areas and optimal for landslides that often cover an area that range from less than one square kilometres up to few square kilometres. Ultra-high resolution images acquired by RPAS can support the definition not only of the identification of studied landslide limit, but also the identification and mapping of main geomorphological features (Rossi et al., 2017; Fiorucci et al., 2018). Furthermore, a sequence of RPAS acquisitions over the time can provide useful support for the study of the gravitational process evolution.

According to Scaioni et al. (2014), applications of remote sensing for landslides investigations can be divided into three classes: i) landside recognition, classification and post-event analysis, ii) landslide monitoring, iii) landslide susceptibility and hazard assessment.

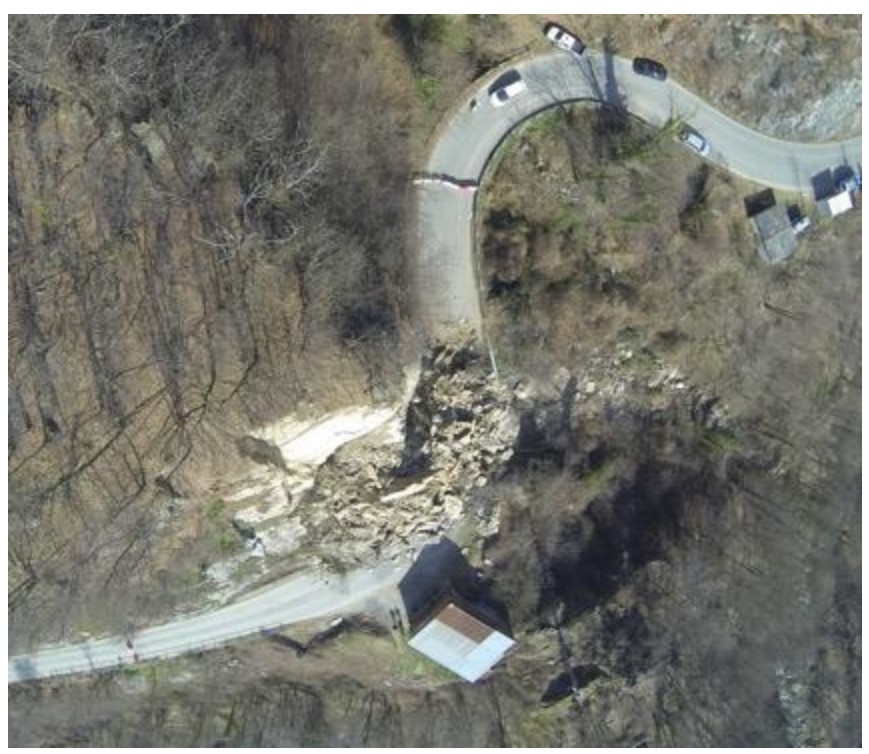


Figure 3. Example of RPAS image of a rockslide occurred on a road. The image was acquired after the
rockslide occurred in 2014 in San Germano municipality (Piemonte region, NW Italy). As presented in
Giordan et al. (2015a), a multi-rotor of local Civil Protection Agency was used to evaluate occurred damages
and residual risk. RPAS images can be very useful to have a representation from a different point of view of
the occurred phenomena. Even not already processed using SFM applications, this dataset can be very
useful for decision makers to define the strategy for the management of the first phase of emergency.

**2.1.1 Landslides recognition**
The identification and mapping of landslides are usually performed after intense meteorological events that
can activate or reactivate several gravitational phenomena. The identification and mapping of landslides can
be organized in landslides event maps. Landslides event mapping is a well-known activity obtained thought
field surveys (Santangelo et al., 2010), visual interpretation of aerial or satellite images (Brardinoni et al.,
2003; Ardizzone et al., 2013) combined analysis of LiDAR DTM and images (Van Den Eeckhaut et al., 2007;
Haneberg et al., 2008; Giordan et al., 2013; Razak et al., 2013; Niculiţă et al., 2016). The use of RPAS for the
identification and mapping of a landslide has been described by several authors (Niethammer et al 2009;
Niethammer et al 2010; Rau et al., 2011; Carvajal et al., 2011; Travelletti et al., 2012; Torrero et al., 2015;
Casagli et al., 2017). Niethammer et al 2009 and Liu et al. (2015) showed how RPAS could be considered a
good solution for the acquisition of ultra-high resolution images with low-cost systems. Fiorucci et al. (2018)
compared the results of the landslide limit mapped using different techniques and found that satellite images
can be considered a good solution for the identification and map of landslides over large areas. On the
contrary, if the target of the study is the definition of landslide's morphological features, the use of more
detailed RPAS images seemed to be the better solution. As suggested by Walter et al., (2009) and Huang et
al., (2017) one of the most critical elements for a correct georeferencing of acquired images are the use of
GCPs. The in situ installation and positioning acquisition of GCPs can be an important challenge in particular
in dangerous areas as active landslides. Very often, GCPs are not installed in the most active part of the slide
but on stable areas. This solution can be safer for the operator, but it can also reduce the accuracy of the
final reconstruction.
Another parameter that can be considered during the planning of the acquisition phase is the morphology of
the studied area. According to with Giordan et al., (2015b), slope materials and gradient can affect the flight
planning and the approach used for the acquisition of the RPAS images. Two possible scenarios can be
identified: i) steep to vertical areas (>40°); ii) slopes with gentle to moderate slopes (<40°). In the first case,
the use of multi-copters with oblique acquisitions is often the best solution. On the contrary, with more
gentle slopes, the use of fixed-wing systems can assure the acquisition of wider areas.

### 2.1.2 Landslides monitoring

The second possible field of application of RPAS is the use of multi-temporal acquisitions for landslides
monitoring. This topic has been described by several authors (Dewitte et al., 2008; Turner and Lucieer, 2013;
Travelletti et al., 2012; Lucieer et al. 2014a; Turner et al., 2015; Marek et al., 2015; Lindner et al., 2016; Peppa
et al., 2017). In these works, numerous techniques based on the multi-temporal comparison of RPAS datasets
for the definition of the evolution of landslides have been presented and discussed. Niethammer et al. (2010
and 2012) described how the position change of geomorphological features (in particular fissures) could be
considered for a multi-temporal analysis with the aim of the characterization of the landslide evolution.
Travelletti et al. (2012) introduced the possibility of a semi-automatic image correlation to improve this
approach. The use of image correlation techniques has been also described by Lucieer et al. (2014a) who
demonstrated that COSI-Corr (Co-registration of Optically Sensed Imaged and Correlation - Leprince et al.
2007, 2008; Ayoub et al., 2009) can be adopted for the definition of the surface movement of the studied
landslide. A possible alternative solution is the multi-temporal analysis of the use of DSMs. The comparison
of digital surface models can be used for the definition of volumetric changes caused by the evolution of the
studied landslide. The acquisition of these digital models can be done with terrestrial laser scanners (Baldo
et al., 2009) or airborne LiDAR (Giordan et al., 2013). Westoby et al. (2012) emphasized the advantages of
RPAS concerning terrestrial laser scanner, which can suffer from line-of-sight issues, and airborne LiDAR,
which are often cost-prohibitive for individual landslide studies. Turner et al. (2015) stressed the importance
of a good co-registration of multi-temporal DSM for good results that could decrease the accuracy of results.
The use of benchmarks in areas not affected by morphological changes can be used for a correct calibration
of rotational and translation parameters.

### 2.1.3 Landslides susceptibility and hazard assessment

Landslides susceptibility and hazard assessment are often performed at basin scale (Guzzetti et al., 2005)
using different remote sensing techniques (Van Westen et al., 2008). The use of RPAS can be considered for
single case study applications to help decision makers in the identification of the landslide damages and the
definition of residual risk (Giordan et al., 2015a). Saroglou et al., (2017) presented the use of RPAS for the
definition of trajectories of rock falls prone areas. Salvini et al. (2017 and 2018) and Török et al., (2017)
described the combined use of TLS and RPAS for hazard assessment of steep rock walls. All these papers
considered the use of RPAS as a valid solution for the acquisition of DSM over sub-vertical areas. Török et al.,
(2017) and Tannant et al., 2017 also described in their manuscripts how RPAS DSMs can be used for the
evaluation of slope stability using numerical modelling. Fan et al. (2017) analyzed the geometrical features

and provided the disaster assessment of a landslide occurred on June 24 2017 in the village of Xinmo in Maoxian County, (Sichuan Province, Southwest China). Aerial images were acquired the day after the event from an unmanned aerial vehicle (UAV) (fixed-wing UAV, with a weight less than 10 kg, and flight autonomy up to 4 hours), and a digital elevation model (DEM) was processed, with the purpose to analyzed the main landslide geometrical features (front, rear edge elevation, accumulation area, horizontal sliding distance)

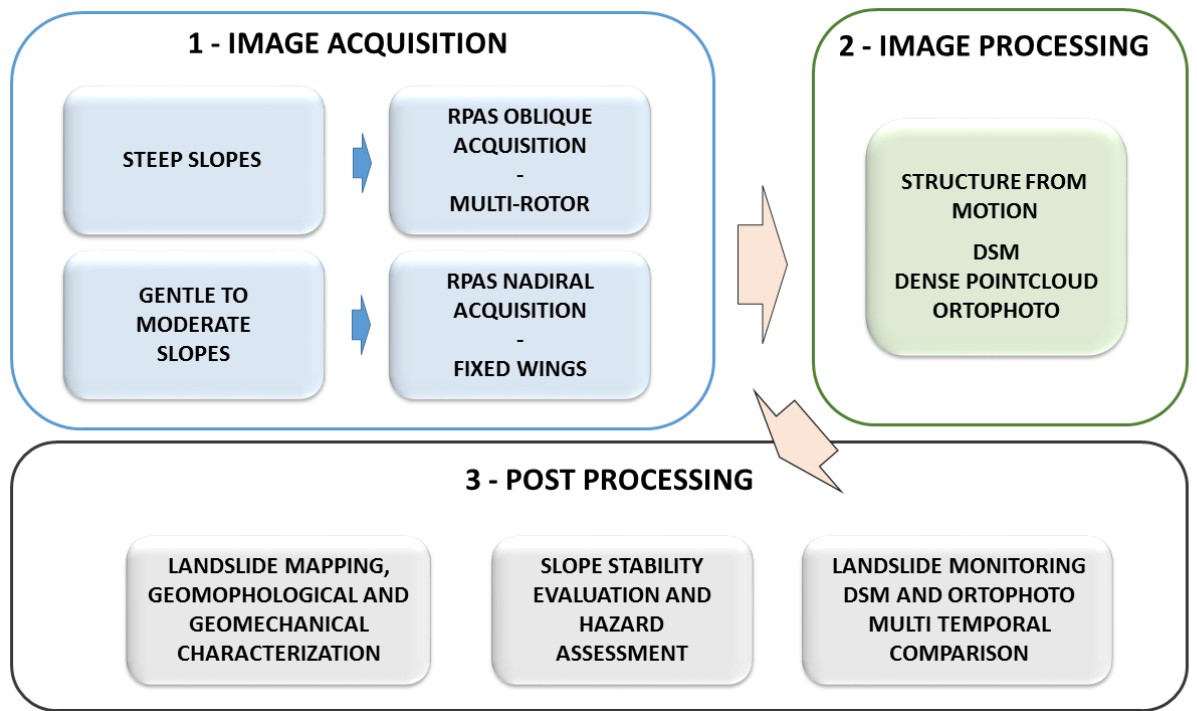

Figure 4. Acquisition, processing and post-processing of RPAS images applied to i) landslides recognition, ii) hazard assessment and iii) slope evolution monitoring.

## 2.2 Floods

Disastrous floods in urban, lowland areas often cause fatalities and severe damage to the infrastructure. Monitoring the flood flow, assessment of the flood inundation areas and related damages, post-flood landscape changes, and pre-flood prediction are therefore seriously required. Among various scales of approaches for flood hazards (Sohn et al., 2008), the RPAS has been adopted for each purpose of the flood damage prevention and mitigation because it has an ability of quick measurement at a low cost (DeBell et al., 2016; Nakamura et al., 2017). Figure 5 shows an example of the use of RPAS for prompt damage assessment by a severe flood occurred on early July 2017 at northern Kyushu area, southwest Japan. The Geospatial Information Authority of Japan (GSI) utilized an RPAS for the post-flood video recording and photogrammetric mapping of the damaged area with flood flow and large woody debris.

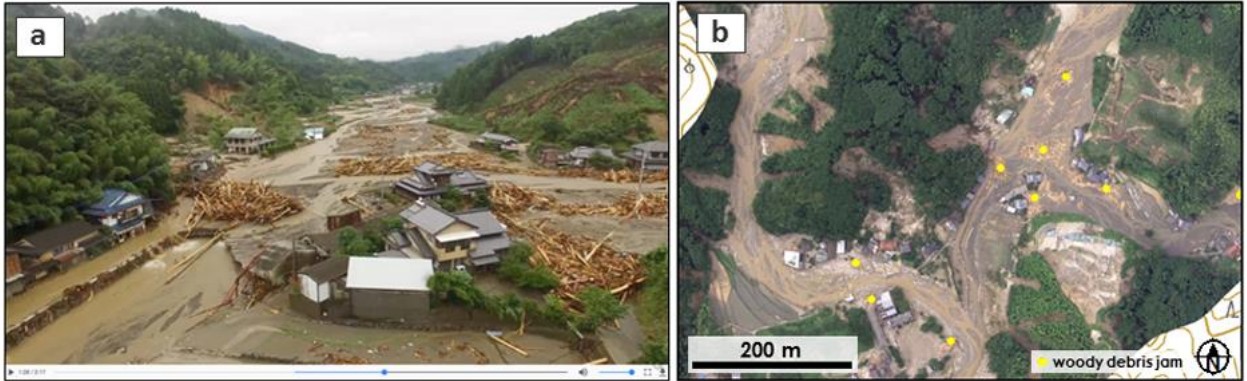


Figure 5. Image captures of flood hazard using RPAS just after the 2017 Northern Kyushu Heavy Rain in the
early July (southwest Japan), provided by GSI. (a) A screenshot of the aerial video of a flooded area along
the Akatani River, Asakura City in Fukuoka Prefecture. (b) Orthorectified image of the damaged area.
Locations of woody debris jam are mapped and shown on the online map (GSI, 2017). The video and map
products are freely provided (compatible with Creative Commons Attribution 4.0 International).

## 2.2.1. Potential analysis of flood inundation

The risk assessments of flood inundation before the occurrence of a flood is crucial for the mitigation of the
flood-disaster damages. RPAS is capable of providing quick and detailed analysis of the land surface
information including topographic, land cover, and land use data, which are often incorporated into the
hydrological modelling for the flood estimate (Costa et al., 2016). As a pre-flood assessment, Li et al. (2012)
explored the area around an earthquake-derived barrier lake using an integrated approach of remote sensing
including RPAS for the hydrological analysis of the potential dam-break flood. They proposed a technical
framework for the real-time evacuation planning by accurately identifying the source water area of the
dammed lake using a RPAS, followed by along-river hydrological computations of inundation potential.
Tokarczyk et al. (2015) showed that the RPAS-derived imagery is useful for the rainfall-runoff modelling for
the risk assessment of floods by mapping detailed land-use information. As a key input data, high-resolution
imperviousness maps were generated for urban areas from RPAS imagery, which improved the hydrological
modelling for the flood assessment. Zazo et al. (2015) and Şerban et al. (2016) demonstrated hydrological
calculations of the potentially flood-prone areas using RPAS-derived 3D models. They utilized 2D cross
profiles derived from the 3D model for the hydrological modelling.

## 2.2.2. Flood monitoring

Monitoring of the ongoing flood is potentially important for the real-time evacuation planning. Le Coz et al.
(2016) mentioned that the movies captured by a RPAS, which can be operated by not only research specialists
but also general non-specialists, is potentially useful for the quantitative monitoring of floods including flow
velocity estimate and flood modelling. This can also contribute to the crowdsourced data collection for flood
hydrology as the citizen science. In case of flood monitoring, however, areas under water is often problematic
by image-based photogrammetry because the bed is not often fully seen in aerial images. If the water is clear
enough, bed images under water can be captured, and the bed morphology can be measured with additional
corrections of refraction (Tamminga et al., 2015; Woodget et al., 2015), but the flood water is often unclear
because of the abundant suspended sediment and disturbing flow current. Another option is the fusion of
different datasets using a sonar-based measurement for the water-covered area, which is registered with
the terrestrial datasets (Flener et al., 2013; Javernick et al., 2014). Image-based topographic data of water
bottom by unmanned underwater vehicle (UUV, also known as an autonomous underwater vehicle, AUV)
can also be another option (e.g., Pyo et al., 2015), although such the application of UUV to flooding has been
limited.
Not only the use of topographic datasets derived from Structure from Motion-Multi Stereo View (SfM-MVS)
photogrammetry, the use of orthorectified images concurrently derived from the RPAS-based aerial images
is advantageous for the assessment of hydrological observation and modelling of floods. Witek et al. (2014)
developed an experimental system to monitor the stream flow in real time for the prediction of overbank
flood inundation. The real-time prediction results are also visualized online with a web map service with a
high-resolution image (3 cm/pix). Feng et al. (2015) reported that the accurate identification of inundated
areas is feasible using RPAS-derived images. In their case, deep learning approaches of the image
classification using optical images and texture by RPAS successfully extracted the inundated areas, which
must be useful for flood monitoring. Erdelj et al. (2017) proposed a system that incorporates multiple RPAS
devices with wireless sensor networks to perform the real-time assessment of a flood disaster. They
discussed the technical strategies for the real-time flood disaster management including the detection,
localization, segmentation, and size evaluation of flooded areas from RPAS-derived aerial images.

**2.2.3. Post-flood changes**
Post-flood assessments of the land surface materials including topography, sediment, and vegetation are
more feasible by RPAS surveys (Izumida et al., 2017). Smith et al. (2014) proposed a methodological
framework for the immediate assessment of flood magnitude and affected landforms by SfM-MVS
photogrammetry using both aerial and ground-based photographs. In this case, it is recommended to
carefully select appropriate platforms for SfM-MVS photogrammetry (either airborne or ground-based)
based on the field conditions. Tamminga et al. (2015) examined the 3D changes in river morphology by an
extreme flood event, revealing that the changes in reach-scale channel patterns of erosion and deposition
are poorly modelled by the 2D hydrodynamics based on the initial condition before the flood. They also
demonstrate that the topographic condition can be more stable after such an extreme flood event.
Langhammer et al. (2017) proposed a method to quantitatively evaluate the grain size distribution using
optical images taken by a RPAS, which is applied to the sediment structure before and after a flash flood.
As a relatively long-term study, Dunford et al. (2009) and Hervouet et al. (2011) explored annual landscape
changes after the flood using RPAS-derived images together with other datasets such as satellite image
archives or a manned motor paraglider. Their work assessed the progressive development of vegetation on
a braided channel at an annual scale, which appears to be controlled by local climate including rainfall,
humidity, and air temperature, hydrology, groundwater level, topography, and seed availability. Changes in
the sediment characteristics by a flood is another key feature to be examined.

## 2.3 Earthquakes
Remote sensing technology has been recognized as a suitable source to provide timely data for automated
detection of damaged buildings for large areas (Dong and Shan, 2013; Pham et al., 2014; Cannioto et al.,
2017). In the post-event, satellite images have been traditionally used for decades to visually detect the

damages on the buildings to prioritize the interventions of rescuers. Operators search for externally visible damage evidence such as spalling, debris, rubble piles and broken elements, which represent strong indicators of severe structural damage. Several researches, however, have demonstrated how this kind of data often leads to the wrong detection, usually underestimating the number of the collapsed building because of their reduced resolution on the ground. In this regard, airborne images and in particular oblique acquisitions (Tu et al., 2017; Nex et al., 2014; Gerke and Kerle 2011; Nedjati et al., 2016) have demonstrated to be a better input for reliable assessments, allowing the development of automated algorithms for this task (Figure 6). The deployment of photogrammetric aeroplanes on the strike area is however very often unfeasible especially when the early (in the immediate hours after the event) damage assessment for response action is needed.

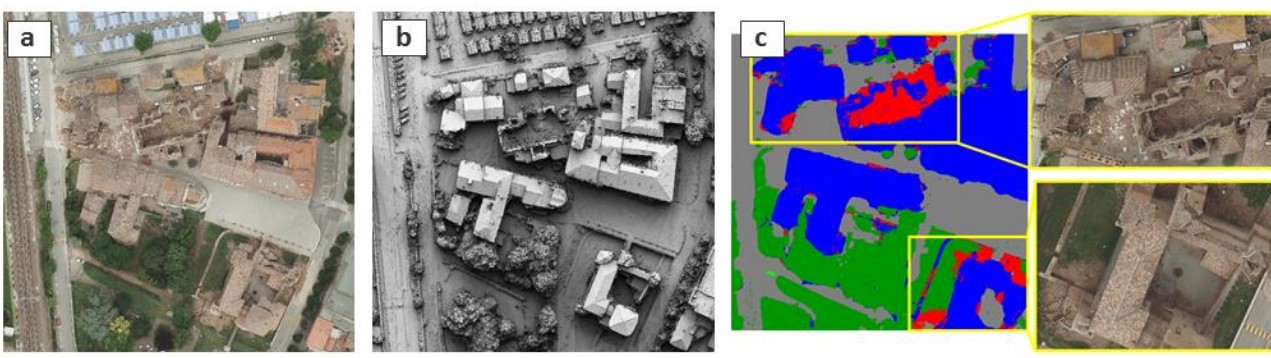

Figure 6. True-orthophoto, Digital Surface Model and damage map of an urban area using airborne nadir images (Source: Nex et al., 2014).

For this reason, RPASs have turned out to be valuable instruments for the building damage assessment (Hirose et al., 2015). The main advantages of RPASs are their availability (and reduced cost) and the ease to repeatedly acquire high-resolution images. Thanks to their high resolution, their use is not only limited to the early impact assessment for supporting rescue operations, but it is also considered in the preliminary analysis of the structural damage assessment.

### 2.3.1 Early impact assessment

The fast deployment in the field, the easiness of use and the capability to provide in real time high-resolution information of inaccessible areas to prioritize the operator's activities are the strongest point of RPASs for these activities (Boccardo et al., 2015). The use of RPASs for rescue operations started almost a decade ago (Bendea et al., 2008) but their massive adoption has begun only in the very last few years (Earthquake in Nepal 2015) thanks to the development of low cost and easy to use platforms. Initiatives like *UAViators* (http://uaviators.org/) have further increased the public awareness and acceptance of this kind of instruments. Several rescue departments have now introduced RPAS as part of the conventional equipment of their teams (Xie et al., 2014). The huge number of videos acquired by RPAS and posted by rescuers online (i.e. Youtube) after the 2016 Italian earthquakes confirm this general trend.

The operators use RPASs to fly over the interest area and get information through visual assessment of the streaming videos. The quality of this analysis is therefore limited to the ability of the operator to fly the RPAS over the interest area. The lack of video geo-referencing usually reduces the interpretability of the scene and the accurate localization of the collapsed parts: only small regions can be acquired in a single flight. The lack

of georeferenced maps prevents the smooth sharing of the collected information with other rescue teams
limiting the practical exploitation of these instruments. RPASs are mainly used in daylight conditions as the
flight during the night is extremely critical, and the use of thermal images is of limited help for the rescuers.
Many researchers have developed algorithms to automatically extract damage information from imagery
(Figure 7). The main focus of these works is to reliably detect damages in a reduced time to satisfy the time
constraints of the rescuers. In (Vetrivel et al., 2015) the combined use of images and photogrammetric point
clouds have shown promising results thanks to a supervised approach. This work, however, highlighted how
the classifier and the designed 2D and 3D features were hardly transferable to different datasets: each scene
needed to be trained independently strongly limiting the efficiency of this approach. In this regard, the recent
developments in machine learning (i.e. Convolutional Neural Networks, CNN) have overcome these limits
(Vetrivel et al., in press), showing how they can correctly classify scenes even if they were trained using other
datasets: a trained classifier can be directly used by rescuers on the acquired images without need for further
operations. The drawback of these techniques is the computational time: the use of CNN, processing like
image segmentation or point cloud generation are computationally demanding and hardly compatible with
real-time needs. In this regard, most recent solutions exploit only images (i.e. no need to generate point
cloud) and limit the use of most expensive processes to the regions where faster classification approaches
provide uncertain results to deliver an almost real-time information (Duarte et al., 2017).

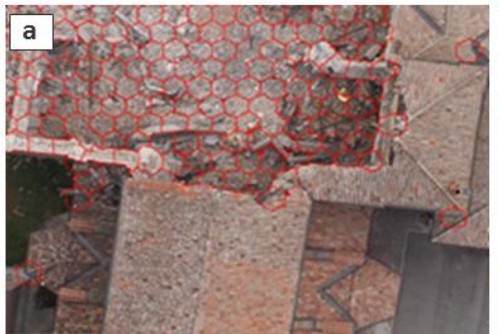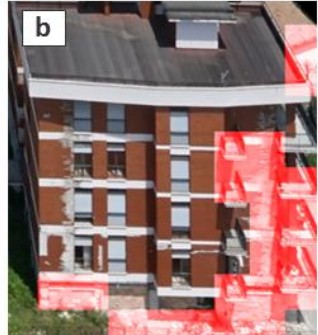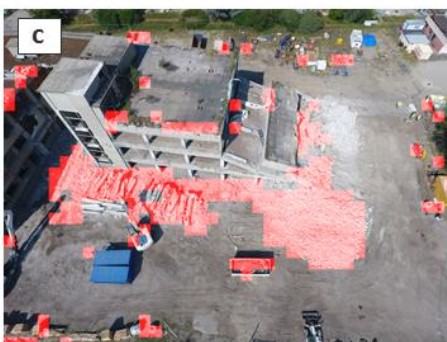


Figure 7. Examples of damage detection on images acquired in three different scenarios (a) Mirabello (source: Vetrivel
et al., in press) and (b) L'Aquila and Lyon (source Duarte et al., 2017).

### 403 2.3.2 Building damage assessment

The damage evidence that can be captured from a UAV is not sufficient to infer the actual damage state of
the building as it requires additional information such as damages to internal building elements (e.g., columns
and beams) that cannot be directly defined from images. Even though this information is limited, images can
provide useful information about the external condition of the structure, evidencing anomalies and damages
and providing a first important information for structural engineers. Two main typologies of investigations
can be performed: (i) the use of images for the detection of cracks or damages on the external surfaces of
the building (i.e. walls and roofs) and (ii) the use of point clouds (generated by photogrammetric approach)
to detect structural anomalies like tilted or deformed surfaces. In both cases, the automated processing can
only support and ease the work of the expert who still interprets and assess the structural integrity of the
building.

In (Fernandez-Galarreta et al., 2015) a comprehensive analysis of both point clouds and images to support the ambiguous classification of damages and their use for damage score was presented. In this paper, the use of point clouds was considered efficient for more serious damages (partial or complete collapse of the building), while images were used to identify smaller damages like cracks that can be used as the basis for the structural engineering analysis. The use of point clouds is investigated in (Baiocchi et al., 2013; Dominici et al., 2017): this contribution highlights how point clouds from UAVs can provide very useful information to detect asymmetries and small deformations of the structure.

## 2.4 Volcanic activity

RPAS is particularly advantageous when the target area of measurement is hardly accessible on the ground due to dangers of volcanic gas or risks of eruption in volcanic areas (Andrews, 2015). Although an equipment of RPAS can be lost or damaged by the volcanic activities, the operator can safely stay in a remote place. Various sensors can be mounted on a RPAS to monitor volcanic activities including topography, land cover, heat, gas composition, and even gravity field (Saiki and Ohba, 2010; Deurloo et al., 2012; Astuti et al., 2009; Middlemiss et al., 2016). The photogrammetric approach to obtain topographic data is widely applied because RGB camera sensors are small enough to be mounted on a small aircraft. As mentioned before, this paper considers in particular small RPAS. In the study of volcanoes, larger aircrafts with a payload of kilograms are also utilized to mount other types of sensors to monitor various aspects of their dynamic activities. For this reason, in this chapter we consider also larger RPAS solutions.

### 2.4.1. Topographic measurements of volcanoes

Long-distance flight of a RPAS enables quick and safe measurements of an emerging volcanic island. Tobita et al. (2014a) successfully performed a fixed-wing RPAS flight for a one-way distance of 130 km in total flight time of 2 hours and 51 minutes over the sea to capture aerial images of a newly formed volcanic island next to Nishinoshima Island (Ogasawara Islands, southwest Pacific). They performed SfM-MVS photogrammetry of the aerial images taken back from the RPAS to generate a 2.5 m resolution DEM of the island. The team also performed two successive measurements of Nishinoshima Island in the following 104 days, revealing the morphological changes in the new island covering a 1,600 m by 1,400 m area (Nakano et al., 2014; Tobita et al., 2014b).

Since the volcanic activities often last for a long period, it is also important to connect the recent volcanic morphological changes to those in the past. Although detailed morphological data of volcanic topography is often unavailable, historical aerial photographs taken in the past decades can be utilized to generate topographic models at a certain resolution. Some case studies have used archival aerial photographs in volcanoes for periods of more than 60 years, generating DEMs with resolutions of several meters for areas of 10 km$^2$ (Gomez, 2014; Derrien et al., 2015; Gomez et al. 2015). Although these DEMs are coarser than those derived from RPAS, they can be used as supportive datasets for the modern morphological monitoring using RPAS at a higher resolution and measurement frequency.

### 2.4.2. Gas monitoring and product sampling

Caltabiano et al. (2005) proposed the architecture of a RPAS for the direct monitoring of gas composition in volcanic clouds of Mt. Etna in Italy. In this system, the 2-m wide fixed-wing RPAS can fly autonomously up to 4000 m altitude with a speed of 40 km/h. Like this system, a RPAS with a payload of several kilograms can carry multiple sensors to monitor different compositions of volcanic gas. McGonigle et al. (2008) used a RPAS for volcanic gas measurements at La Fossa crater of Mt. Vulcano in Italy. The RPAS has 3 kg payload and allows to host an ultraviolet spectrometer, an infrared spectrometer, and an electrochemical sensor on board. The combination of these sensors enabled the estimation of the flux of $SO_2$ and $CO_2$, which are crucial for revealing the geochemical condition of erupting volcanoes. The monitoring of gas composition including $CO_2$, $SO_2$, $H_2S$, $H_2$, as well as the air temperature, can be used for the quantification of the degassing activities and prediction of the conduit magma convection, as suggested by the tests at several volcanoes in Japan (Shinohara, 2013; Mori et al., 2014) and in Costa Rica (Diaz et al., 2015).

A RPAS can also transport a small ground-running robot (Unmanned Ground Vehicle: UGV) to slope head of an active volcano, where the UGV takes close-range photographs of volcanic ash on the ground surface by running down the slope (Nagatani et al., 2013). Protocols for direct sampling of volcanic products using a RPAS have also been developed (Yajima et al., 2014).

### 2.4.3. Geothermal monitoring

In New Zealand, Harvey et al. (2016) and Nishar et al. (2016) carried out experimental studies on the regular monitoring of intense geothermal environments using a small RPAS. They used thermal images taken by an infrared imaging sensor together with normal RGB images for photogrammetry, mapping both the ground surface temperature with detailed topography and land cover data. Chio and Lin (2017) further assessed the use of a RPAS equipped with a thermal infrared sensor for the high-resolution geothermal image mapping in a volcanic area in Taiwan. They improved the measurement accuracies using an onboard sensor capable of post-processed kinematic GNSS positioning. This allows accurate mapping with less ground control points, which are hard to place on such intense geothermal fields.

## 2.5 Wildfires

Wildfires are a phenomenon with local and global effects (Filizzola et al., 2017). Wildfires represent a serious threat for land managers and property owners; in the last few years, this threat has significantly expanded (Peters et al., 2013). The literature also suggests that climate change will continue to enhance the potential forest fire activity in different regions of the world (McKenzie et al. 2014; Abatzoglou and Williams, 2016). Remote sensing technologies can be very useful in monitoring such hazard (Shroeder et al., 2016). Several scientists in the last few years used satellites in fire monitoring (Shroeder et al., 2016). More recently, RPASs have been considered to be useful as well (Martinez-de Dios et al., 2011). Hinkley and Zajkowski (2011) presented the results of a collaborative partnership between NASA, and the US Forest Service established for testing thermal image data for wildfires monitoring. A small unmanned airborne system served as a sensor platform. The outcome was an improved tool for wildfire decision support systems. Merino et al. (2012) described a system for forest fire monitoring using a RPAS. The system integrates the information from the fleet of different vehicles to estimate the evolution of the forest fire in real time. The field tests indicated that RPAS could be very helpful for the activities of firefighting (e.g. monitoring). Indeed, they cover the gap between the spatial scales given by satellites and those based on cameras. Wing et al. (2014) underlined the fact that spectral and thermal sensors mounted in RPASs may hold great promise for future remote sensing

applications related to forest fires. RPASs have greater potential to provide enhanced flexibility for
positioning and repeated data collection. Tang and Shao (2015) summarize various approaches of remote
drone sensing to surveying forests, mapping canopy gaps, measuring forest canopy height, tracking forest
wildfires, and supporting intensive forest management. These authors underlined the usefulness in using
drones for wildfire monitoring. RPASs can repeatedly fly to record the extent of an ongoing wildfire without
jeopardizing crews' safety. Zajkowski et al. (2015) tested different RPASs (e.g. quadcopter, fixed-wing) for the
analysis of fire activity. Measurements included visible and long-wave infrared (LWIR) imagery, black carbon,
air temperature, relative humidity and three-dimensional wind speed and direction. The authors also
described in detail the mission's plan, including the logistics of integrating RPAS into a complex operations
environment, specifications of the aircraft and their measurements, execution of the missions and
considerations for future missions. Allison et al. (2016) provided a detailed state of the art on fire detection
using both manned and unmanned aerial platforms. This review highlighted the following challenges: the
need to development of robust automatic detection algorithms, the integration of sensors of varying
capabilities and modalities, the development of best practices for the use of new sensor platforms (e.g. mini
RPAS), and their safe and effective operation in the airspace around a fire.

## 3. Discussion and conclusion

In this paper, we analysed possible applications of RPAS to natural hazards. The available literature on this
topic is strongly increased in last few years, according to the improvement of the diffusion of these systems.
In particular, we considered: landslides, floods, earthquakes, volcanic activities and wildfires.
RPAS can support studies on active geological processes and can be considered a good solution for the
identification of effects and damages due to several catastrophic events. One of the most important elements
that characterized the use of RPAS is their flexibility and versatility, largely confirmed by the wide number of
operative solutions available in the literature. The available literature pointed out the necessity of the
development of dedicated methodologies that can be able to take the full advantage of RPAS. In particular,
typical results of structure from motion software (orthophoto and DSM) that are considered the end of
standard data-processing, can be very often the starting point of dedicated procedures specifically conceived
for natural hazards applications.
In the pre-emergency phase, one of the main advantages of RPAS surveys is to acquire high resolution and
low-cost data to analyse and interpret environmental characteristics and potential triggering factors (e.g.
slope, lithology, geostructure, land use/land cover, rock anomalies, and displacement). The data can be
collected with high revisit times to obtain multi-temporal observations. After the characterization of hazard
potential and vulnerability, some areas can be identified by a higher level of risk. These cases request an
intensive monitoring, to gain a quantitative evaluation of the potential occurrence of an event. In this
context, the use of aerial data represents a very useful complementary data source concerning the
information acquired through ground-based observations in particular for dangerous areas.
During the emergency phase, high-resolution imagery is asked to be acquired over the event site. The primary
use of this data is for the assessment of the damage grade (extent, type and damage grades specific to the
event and eventually of its evolution). They may also provide relevant information that is specific to critical
infrastructures, transport systems, aid and reconstruction logistics, government and community buildings,
hazard exposure, displaced population, etc (Ezequiel et al., 2014). Concurrently, the availability of clear and
straightforward raster and vector data, integrated with base cartographic contents (transportation, surface

hydrology, boundaries, etc.) it is recognized as an added-value to support decision makers for the management of emergency operations (Fikar et al., 2016). These applications very often need prompt and reliable interventions. RPAS should, therefore, deliver information promptly. In this regard, very few researchers have focused on this issue: most of the reported works present (often time-consuming and even manual) post-processing of the acquired data, precluding the use of their results from practical and real-life scenarios. A big effort should be taken by the research community to propose faster and automated approaches. In particular during emergencies, the time required for RPAS dataset processing is an important element that should be carefully considered. Giordan et al. (2015a) presented a case study related to a landslide emergency. In this paper, authors considered not only possible results but also the time that is required for them

As in many other domains, RPAS present a disruptive technology where, beside conventional SfM applications for 3D reconstructions, many dedicated and advanced methodologies are still in their experimental phase and will need to be further developed in the incoming years. In the following years, it would be desirable to witness the transfer of the best practices in the use of RPAS be then from the Research community to Government Agencies (or private companies) involved in the prevention and reduction of impacts of natural hazards. The Scientific community should contribute to the definition of standard methodologies that can be assumed by civil protection agencies for the management of emergencies.

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
