# Peer review of "REVIEW ARTICLE: THE USE OF REMOTELY PILOTED AIRCRAFT SYSTEMS (RPAS) FOR NATURAL HAZARDS MONITORING AND MANAGEMENT"

_Natural Hazards and Earth System Sciences, 2017_

## Referee Comment (RC1) · Anonymous Referee #1 · 26 Nov 2017

1. Overall This is a review paper relating to the use of RPAS for natural hazard monitoring and management. It particularly focusing on the use of Mini and Micro RPAS for five kinds of disaster, such as landslides, floods, earthquakes, wildfires and volcano activities. However, the topic and discussed disaster types are similar to the following paper just published last year. Thus, I suggest to major revise this manuscript. Detail comments are stated below. ïĄň Christopher Gomez and Heather Purdie, 2016, "UAV-based Photogrammetry and Geocomputing for Hazards and Disaster Risk Monitoring – A Review", Geoenvironmental Disasters, Vol.3, No.23. 2. Comments i. The above mentioned article was not referenced, compared or analyzed. It is strongly suggest to include this paper and conduct comparisons to emphasize their different point of view.

[Figure]

ii. The used acronyms are not consistent, RPAS, UAV, UAS, UVS were adopted at different places of the paper. If their definitions have major difference, the authors should define them clearly. If not, using one acronym for the whole paper may be considered. iii. Line 28-31, numbers within () should include unit, such as 380, 22765, etc. iv. Line 48, what is RLS and what are RTK/PPK at Line 95? The first time an acronym appear, its whole name should be explained. On the contrary, the explanation of GCP appear twice in the paper. v. Table 1 specify the classification of Mini/Micro UAV. A reference should be referred. vi. Line 476, "small UAV" is used. What is its definition? vii. Meanwhile, I doubt the definition in Table 1 is correct, as the Max. Flight altitude for Micro UAV is FIXED at 250m and its endurance time is also FIXED at 1h. viii. In this paper, the authors focus on the use of Mini and Micro RPAS only. However, these two kinds of RPAS are not suitable for volcano activities study, because its maximum flight altitude is generally lower than a volcano. For example at Line 422, a fixed-wing UAV can fly over Mt. Etna up to 4000m. This fixed-wing is not belong to the Mini or Micro RPAS. Right? There are other similar case studies that didn't use Mini or Micro RPAS as well. ix. Line 212, RPAs or RPAS? x. Line 415, two references for Gomez are not found in the list of reference.

---

## Author Comment (AC1) · 19 Dec 2017

Answer to Referee comments 1

RC - Referee comments AC- Author comments

RC - Overall This is a review paper relating to the use of RPAS for natural hazard monitoring and management. It particularly focusing on the use of Mini and Micro RPAS for five kinds of disaster, such as landslides, floods, earthquakes, wildfires and volcano activities. However, the topic and discussed disaster types are similar to the following paper just published last year. Thus, I suggest to major revise this manuscript.

AC - We would like to thank the Reviewer for his suggestions. We well know that the topic has also been analyzed by other authors. However, with our review, we tried to make the literature review more complete and updated. We provided more than 150 references considering the most important natural hazards, including some recent articles published in the last year and following the available Annual Disaster Statistical Review. Along these lines, we believe that the natural hazards scientific community will be benefited by such long and updated list of articles and research advances. In detail, the manuscript is focused on the revision of available bibliography for the use of RPAS for: landslides, earthquakes, volcanic activity and wildfires. These four categories are the most dangerous and the manuscript propose a revision of case studies and proposed methodology to fix a possible approach for the use of RPAS in these critical conditions. The description of case studies and possible approaches is important to fix a common methodology that can be used not only for scientific purposes, but also for the management of real emergencies. Until now, the lack of a well-defined methodology that describes pros and cons to the use of RPAS for the support during natural hazard emergencies is a critical aspect that this paper can try to solve.

RC - Detail comments are stated below. ï ËŻAn Christopher Gomez and Heather Purdie, 2016, "UAV- ËĞ based Photogrammetry and Geocomputing for Hazards and Disaster Risk Monitoring – A Review", Geoenvironmental Disasters, Vol.3, No.23. 2. Comments i. The above mentioned article was not referenced, compared or analyzed. It is strongly suggest to include this paper and conduct comparisons to emphasize their different point of view.

AC - Line 132: Done. We would like to thank the Reviewer for this suggestion. We added this important paper in our review, and we also considered the bibliography of the manuscript. In our manuscript, we revised 151 papers (92 more than the paper mentioned above) that can be considered an exhaustive representation of available bibliography on this subject. With respect to Gomez and Purdie (2016), we tried to analyze more in-depth the bibliography and define a possible methodology for the use

of RPAS that we obtained merging all the revised papers. In the case of landslides, for example, this approach has been used to define the possible use of RPAS for: i) landside recognition, classification and post-event analysis, ii) landslide monitoring, iii) landslide susceptibility and hazard assessment. For each of this points, we present a description of the use of RPAS based on the available bibliography. To clarify this point, we add the following paragraph that introduces the paper of Gomez and Purdie and points out the different approach of our manuscript: "Gomez and Purdie (2016) published a detailed analysis of the use of RPAS for hazards and disaster risk monitoring. In our paper, we focused our attention on the most dangerous natural hazards that can be analyzed using RPAS. According to the definitions used by Annual Disaster Statistical Review (Guha-Sapir et al., 2016), the paper considers in particular: i) landslides, ii) floods iii) earthquakes v) volcanic activity vi) wildfires. For each considered category of natural hazard, our paper presents a review of a large list of published papers (151 papers) analyzing proposed methodologies and provided results, and underlining strengths and limitations in the use of RPAS. The aim of this paper is the description of the possible use of RPAS in considered natural hazard, describing a general methodology for the use of these systems in different contexts merging all previously published experiences." In this revised version of the paper, we also added the following bibliography: Derrien, A., Villeneuve, N., Peltier, A. and Beauducel, F.: Retrieving 65 years of volcano summit deformation from multitemporal structure from motion: The case of Piton de la Fournaise (La Réunion Island), Geophys. Res. Lett., 42(17), 6959–6966, doi:10.1002/2015GL064820, 2015. Dewitte, O., J.C. Jasselette, Y. Cornet, M. Van Den Eeckhaut, A. Collignon, J. Poesen, and A. Demoulin.: Tracking landslide displacements by multitemporal DTMs: A combined aerial stereophotogrammetric and LIDAR approach in western Belgium. Engineering Geology, 7, 582–586, 2008. Diaz, J. A., Pieri, D., Wright, K., Sorensen, P., Kline-Shoder, R., Arkin, C. R., Fladeland, M., Bland, G., Buongiorno, M. F., Ramirez, C., Corrales, E., Alan, A., Alegria, O., Diaz, D. and Linick, J.: Unmanned Aerial Mass Spectrometer Systems for In-Situ Volcanic Plume Analysis, J. Am. Soc. Mass Spectrom., 26(2), 292–304, doi:10.1007/s13361-

014-1058-x, 2015. Eugster, H. and Nebiker, S.: UAV-based augmented monitoring–real-time georeferencing and integration of video imagery with virtual globes. In: Int. Archives of Photogrammetry, Remote Sensing and Spatial Information Sciences, Beijing, China, 37(B1), 1229–1235. 2008. Ezequiel, C.A.F., Cua, M., Libatiquem, N.C., Tangonan, G.L., Alampay, R., Labuguen, R.T., Favila, C.M., Honrado, J.L.E., Canos, V., Devaney, C., Loreto, L.B., Bacusmo, J. and Palma, B.: UAV Aerial Imaging Applications for Post-Disaster Assessment, Environmental Management and Infrastructure Development. 2014 International Conference on Unmanned Aircraft Systems (ICUAS) Orlando, Fl, USA proceedings: 274–283, 2014. Fan, J., Zhang, X., Su, F., Ge, Y., Tarolli, P., Yang, Z., Zeng, C., and Zeng, Z.: Geometrical feature analysis and disaster assessment of the Xinmo landslide based on remote sensing data, Journal of Mountain Science, 14, 1677–1688, doi:10.1007/s11629-017-4633-3, 2017. Fikar, C., Gronalt, M., and Hirsch. P.A.: decision support system for coordinated disaster relief distribution. Exp. Syst. Appl. 57, 104–116. doi:10.1016/j.eswa.2016.03.039, 2016. Flener, C., Vaaja, M., Jaakkola, A., Krooks, A., Kaartinen, H., Kukko, A., Kasvi, E., Hyyppä, H., Hyyppä, J. and Alho, P.: Seamless mapping of river channels at high resolution using mobile liDAR and UAV-photography, Remote Sens., 5(12), 6382–6407, doi:10.3390/rs5126382, 2013. Gomez, C.: Digital photogrammetry and GIS-based analysis of the bio-geomorphological evolution of Sakurajima Volcano, diachronic analysis from 1947 to 2006, J. Volcanol. Geotherm. Res., 280, 1–13, doi:10.1016/j.jvolgeores.2014.04.015, 2014. Gomez, C. and Kato, A.: Multi-scale voxel-based algorithm for UAV-derived point-clouds of complex surfaces. IEEE International ICARES – Aerospace Electornics and Remote Sensing Technology: 205–209. 2014. Gomez, C. and Purdie, H.: UAV- based Photogrammetry and Geocomputing for Hazards and Disaster Risk Monitoring – A Review. Geoenvironmental Disasters 3(23), 1-11, 2016. Gomez, C., Hayakawa, Y. and Obanawa, H.: A study of Japanese landscapes using structure from motion derived DSMs and DEMs based on historical aerial photographs: New opportunities for vegetation monitoring and diachronic geomorphology, Geomorphology, 242, 11–20, doi:10.1016/j.geomorph.2015.02.021,

2015. Guha-Sapir, D., Hoyois, P., and Below, R.: Annual Disaster Statistical Review 2015 The numbers and trends. Centre for Research on the Epidemiology of Disasters, Ciaco Imprimerie, Louvain-la-Neuve (Belgium), pp. 50, 2016. Hervouet, A., Dunford, R., Piégay, H., Belletti, B. and Trémélo, M.-L.: Analysis of Post-flood Recruitment Patterns in Braided-Channel Rivers at Multiple Scales Based on an Image Series Collected by Unmanned Aerial Vehicles, Ultra-light Aerial Vehicles, and Satellites, GIScience Remote Sens., 48(1), 50–73, doi:10.2747/1548-1603.48.1.50, 2011. Javernick, L., Brasington, J. and Caruso, B.: Modeling the topography of shallow braided rivers using Structure-from-Motion photogrammetry, Geomorphology, 213, 166–182, doi:10.1016/j.geomorph.2014.01.006, 2014. Lindner, G., Schraml, K., Mansberger, R. and Hubl J.: UAV monitoring and documentation of a large landslide. Appl Geomat, 8(1), 1-11, 2016. Liu, C.-C., Chen, P.-L., Tomoya, M., Chen, C.-Y.: Rapidly responding to landslides and debris flow events using a low-cost unmanned aerial vehicle. J. Rem. Sens. 9(1), 1-11, doi:10.1117/1.JRS.9.096016, 2015. Mori, T., Hashimoto, T., Terada, A., Yoshimoto, M., Kazahaya, R., Shinohara, H. and Tanaka, R.: Volcanic plume measurements using a UAV for the 2014 Mt. Ontake eruption, Earth, Planets Sp., 68(1), 49, doi:10.1186/s40623-016-0418-0, 2016. Nakamura, F., Shimatani, Y., Nishihiro, J., Ohtsuki, K., Itsukushima, R. and Yamada, H.: Report on flood disaster in Kinu River, occurred in September, 2015 (in Japanese with English abstract), Ecol. Civ. Eng., 19(2), 259–267, doi:10.3825/ece.19.259, 2017. Nedjati, A., Vizvari, B., Izbirak, G.: Post-earthquake response by small UAV helicopters, Nat. Hazards 80, 1669–1688, 2016. Doi: http://dx.doi.org/10.1007/ s11069-015-2046-6 Obanawa, H., Y. Hayakawa, and C. Gomez.: 3D Modelling of inaccessible Areas using UAV-based Aerial Photography and Structure from Motion. Transactions of the Japanese Geomorphological Union, 35, 283–294. 2014. Pham, T.-T.-H., P. Apparicio, C. Gomez, C. Weber, and D. Mathon.: Towards a rapid automatic detection of building damage using remote sensing for disaster management. The Haiti earthquake. Dis. Prev. Manage, 23, 53–66, 2014. doi: 10.1108/DPM-12-2012-0148 Pyo, J., Cho, H., Joe, H., Ura, T. and Yu, S.: Development of hovering type AUV " Cyclops " and its performance evaluation using

image mosaicing, Ocean Eng., 109, 517–530, doi:10.1016/j.oceaneng.2015.09.023, 2015. Smith, M. W., Carrivick, J. L., Hooke, J. and Kirkby, M. J.: Reconstructing flash flood magnitudes using "Structure-from-Motion": A rapid assessment tool, J. Hydrol., 519, 1914–1927, doi:10.1016/j.jhydrol.2014.09.078, 2014. Thamm, H.P. and Judex, M.: The "Low cost drone" – An interesting tool for process monitoring in a high spatial and temporal resolution. The International Archives of Photogrammetry, Remote Sensing and Spatial Information Sciences, Enschede, The Netherlands, Vol. XXXVI part 7. 2006 Tamminga, A. D., Eaton, B. C. and Hugenholtz, C. H.: UAS-based remote sensing of fluvial change following an extreme flood event, Earth Surf. Process. Landforms, 40(11), 1464–1476, doi:10.1002/esp.3728, 2015. Witek, M., Jeziorska, J. and Niedzielski, T.: An experimental approach to verifying prognoses of floods using an unmanned aerial vehicle, Meteorol. Hydrol. Water Manag., 2(1), 3–11 [online] Available from: http://www.mhwm.pl/An-experimantal-approach-to-verifying-prognoses-of-floods-using-unmanned-aerial-vehicle,0,8.html, 2014. Woodget, A. S., Carbonneau, P. E., Visser, F. and Maddock, I. P.: Quantifying submerged fluvial topography using hyperspatial resolution UAS imagery and structure from motion photogrammetry, Earth Surf. Process. Landforms, 40, 47–64, doi:10.1002/esp.3613, 2015. Xie, Z., J. Yang, C. Peng, Y. Wu, X. Jiang, R. Li, Y. Zheng, Y. Gao, S. Liu, and B. Tian.: Development of an UAS for post-earthquake disaster surveying and its application in Ms7.0 Lushan Earthquake, Sichuan, China. Comput. Geosc. 68, 22–30, 2014.

RC - ii. The used acronyms are not consistent, RPAS, UAV, UAS, UVS were adopted at different places of the paper. If their definitions have major difference, the authors should define them clearly. If not, using one acronym for the whole paper may be considered.

AC - We revised the text, and we corrected these discrepancies.

RC - iii. Line 28-31, numbers within () should include unit, such as 380, 22765, etc. iv. Line 48, what is RLS and what are RTK/PPK at Line 95? The first time an acronym appear, its whole name should be explained. On the contrary, the explanation of GCP

appear twice in the paper.

AC - We improve the text according to reviewer's suggestions. In particular: 380 are the average number of events per year (we add events in the text) 22,765 are fatalities (we add in the text) RLS should read SLR (single lens reflex) camera (we correct the text) RTK is Real Time Kinematic whereas PPK is Post Processing Kinematic (we add in the text)

RC - v. Table 1 specify the classification of Mini/Micro UAV. A reference should be referred.

AC - UVS International definition, added

RC - vi. Line 476, "small UAV" is used. What is its definition?

AC - fixed

RC - vii. Meanwhile, I doubt the definition in Table 1 is correct, as the Max. Flight altitude for Micro UAV is FIXED at 250m and its endurance time is also FIXED at 1h.

AC - Flight altitude depends on countries whereas endurance depends on the payload. Reported numbers are just indicative.

RC - viii. In this paper, the authors focus on the use of Mini and Micro RPAS only. However, these two kinds of RPAS are not suitable for volcano activities study, because its maximum flight altitude is generally lower than a volcano. For example at Line 422, a fixed-wing UAV can fly over Mt. Etna up to 4000m. This fixed-wing is not belong to the Mini or Micro RPAS. Right? There are other similar case studies that didn't use Mini or Micro RPAS as well.

AC - We thank the reviewer for this important issue. We added in the text that, for volcanoes, we also considered larger RPAS: Line 409: "As mentioned before, this paper considers in particular small RPAS. In the study of volcanoes, larger aircrafts with a payload of kilograms are also utilized to mount other types of sensors to monitor

various aspects of their dynamic activities. For this reason, in this chapter, we also consider larger RPAS solutions."

RC - ix. Line 212, RPAs or RPAS?.

AC - RPAS

RC - x. Line 415, two references for Gomez are not found in the list of reference.

AC - We added missing references, and we made a cross-check of all references mentioned in the manuscript and published in the reference list

---

## Referee Comment (RC2) · Anonymous Referee #2 · 22 Jan 2018

Overall This is a review paper relating to the use of small RPAS for natural hazards monitoring and management for five kinds of disasters, such as landslides, floods, earthquakes, wildfires and volcanos. The paper recites many international papers and summarizes their content and results briefly. The focus is on the use of small RPAS (<30kg MTOW) in combination with optical sensor systems (mainly), laser scanners and gas detection systems. The introduction explains the two classes of RPAS and the common workflow of using an RPAS and post post-processing the aerial single images or video streams (nadir and oblique view) by using common Structure from Motion Software Tools (like Pix4, AgiSoft, Capturing Reality, DroneDeploy, etc.) to generate data products like orthophotos and point clouds. The advantages of using RPAS for natural

hazards assessment are well described related to the use of aerial camera systems (for RGB, Multi-/Hyperspectral and TIR range). Possible accuracies of these data products are described too in dependence of using GCPs, a low cost AHRS and/or high end GNSS/INS system in combination with the optical sensor system. This paper is a good introduction to the usage of RPAS for natural hazards monitoring and even latest results are listed - i.e. using deep learning algorithms / CNN for detecting destroyed facades to provide relevant information on-site and in near realtime for first responders (section 2.3). Sadly, there are no recommendations for best practices or open source tools and no comparison or rating of the described workflows of each section (landslides, floods, earthquakes, wildfires, volcanos). Especially for using SfM-Software many publications are available which analyses image processing time, achievable accuracies of resulting data products by using / not using GCPs, alternating flight strips and/or cross strips and AHRS or GNSS/INS solutions and the effects of using a metric or non-metric camera system - i.e. DJI Phantom 4 Pro (metric) and DJI Mavic (sadly not metric).

Comments Line No. 27: You cite the Annual Disaster Statistical Review of 2015. The Citation ADSR, 2015 is missing in the reference section and I suggest to update the statistic numbers by using the latest report of 2016. Line No. 37: You address a crucial point here. Time matters, especially during the disaster assessment or disaster monitoring phase. With a RPAS you are easily able to monitor on-site in real time. Why is there no section in your paper where you discuss reliable or suitable RPAS solutions compared to common satellite based solutions / services. There is also another issue to be mentioned. Capturing high res images or videos can be done on time but the main bottleneck is the time which is necessary to post-process that huge amount of images (i.e. with SfM Tools) to generate maps, mosaics, orthophotos, point clouds etc. Several case studies have been published by http://drones.fsd.ch/en/ which should be considered to take into account. Line No. 45: "contest" or "context" of remote sensing research? Line No. 48: SLR instead of RLS. I suggest to replace by "integrated camera systems" as well to address all kind of optical solutions for RPAS (i.e. bridge cameras, industrial grade cameras, video cameras, etc.). Section from Line No. 52 to

62: I recommend to add the advantage of "micro RPAS are easy to transport into the disaster area". Foldable Systems (like DJI Mavic) fits easily into a day pack and can be transported safely as hand luggage. Weight matters especially for first responder teams like UNDAC or similar. Section from line no. 83 to 104: I recommend to add some references to papers which analyses possible accuracies by using / not using GCPs and SFM Tools (i.e. Pix4D, Agisoft) or common photogrammetric workflows (i.e. Inpho Match AT). I suggest as well to add some references here to fast mosaicking methods - i.e. PhaseOne and IGI showed promising results with the commercial IGI Mapper System and the German Aerospace Center developed specialized solutions for realtime traffic management (VABENE) and realtime mapping applications (MACS) on manned and unmanned aircrafts. Intro section in general: You name laser scanning and gas detection and also reference on that in section 2.1.1, 2.1.2, 2.1.3, 2.4.2 and 2.5 but a workflow description is missing. I recommend to add this workflow description or to specify the argumentation of using optical sensor systems. Line No. 128: Reference of (ADSR 2015) is missing. Update to ADSR 2016 is recommended. Section 2.1: I recommend to add the main parameters which influence the accuracy of derived DEM and orthophotos (i.e. real GSD, knowledge about interior and exterior orientation parameters, overlap of images, flight strip configuration and used SfM-Software) Line No. 281: First use of SfM-MVS - please explain.

---

## Author Comment (AC2) · 9 Feb 2018

Daniele Giordan et al. daniele.giordan@irpi.cnr.it

ANSWER TO REVIEWER 2

RC - Referee comments AC- Author comments

RC - Overall This is a review paper relating to the use of small RPAS for natural hazards monitoring and management for five kinds of disasters, such as landslides, floods,

earthquakes, wildfires and volcanos. The paper recites many international papers and summarizes their content and results briefly. The focus is on the use of small RPAS (<30 kg MTOW) in combination with optical sensor systems (mainly), laser scanners and gas detection systems. The introduction explains the two classes of RPAS and the common workflow of using an RPAS and post post-processing the aerial single images or video streams (nadir and oblique view) by using common Structure from Motion Software Tools (like Pix4, AgiSoft, Capturing Reality, DroneDeploy, etc.) to generate data products like orthophotos and point clouds. The advantages of using RPAS for natural hazards assessment are well described related to the use of aerial camera systems (for RGB, Multi-/Hyperspectral and TIR range). Possible accuracies of these data products are described too in dependence of using GCPs, a low cost AHRS and/or high end GNSS/INS system in combination with the optical sensor system. This paper is a good introduction to the usage of RPAS for natural hazards monitoring and even latest results are listed - i.e. using deep learning algorithms / CNN for detecting destroyed facades to provide relevant information on-site and in near real time for first responders (section 2.3).

AC – we would like to thank the Reviewer for the good description of the paper that shows several important issues considered.

RC - Sadly, there are no recommendations for best practices or open source tools and no comparison or rating of the described workflows of each section (landslides, floods, earthquakes, wildfires, volcanos). Especially for using SfM-Software many publications are available which analyses image processing time, achievable accuracies of resulting data products by using / not using GCPs, alternating flight strips and/or cross strips and AHRS or GNSS/INS solutions and the effects of using a metric or non-metric camera system - i.e. DJI Phantom 4 Pro (metric) and DJI Mavic (sadly not metric).

AC – We thank the reviewer for this suggestion. In this paper, we decided to focus our attention on natural hazards and possible use of RPAS. The analysis of available bibliography shows that the possible solutions are so different and dependent from the

final goal of the mission and the end users requirements that is quite impossible to propose a generic workflow for each natural hazard. For this reason, we decided to propose a generic workflow in chapter one (figure 2) and then propose a large analysis of available bibliography for each analyzed natural hazard. We also decided to do not compare software or RPAS performances because it was not the aim of the paper and we don't consider the comparison of available software a burning research topic as similar papers have been already published in the past (see Remondino et al., 2014 in Photogrammetric Record). We followed the requests of reviewer 2 and we added several sentences, in particular: From line 106 to line 112: "The use of GCP and different GNSS solutions is a fundamental point. Gerke and Przybilla (2016) presented the effect of RTK-GNSS and cross flight patterns, and Nocerino et al., (2013) presented an evaluation about RPAS processing results quality considering: i) the use of GCPs, ii) different photogrammetric procedures, iii) different network configurations. If a quick mapping is needed, the information delivered by the navigation system can be directly used to stitch the images and produce a rough image mosaicking (Chang-chun et al., 2011)."

RC - Comments Line No. 27: You cite the Annual Disaster Statistical Review of 2015. The Citation ADSR, 2015 is missing in the reference section and I suggest to update the statistic numbers by using the latest report of 2016.

AC – At the moment of submission, ADSR 2016 was not available. Now we updated with this publication. ADSR 2015 was already cited in bibliography as now ADSR 2016, with the suggested citation: Guha-Sapir, D., Hoyois, P., Wallemacq P. and Below, R.: Annual Disaster Statistical Review 2016 The numbers and trends. Centre for Research on the Epidemiology of Disasters, Ciaco Imprimerie, Louvain-la-Neuve (Belgium), pp. 91, 2017

RC - Line No. 37: You address a crucial point here. Time matters, especially during the disaster assessment or disaster monitoring phase. With a RPAS you are easily able to monitor on-site in real time. Why is there no section in your paper where you discuss

reliable or suitable RPAS solutions compared to common satellite based solutions / services. There is also another issue to be mentioned. Capturing high res images or videos can be done on time but the main bottleneck is the time which is necessary to post-process that huge amount of images (i.e. with SfM Tools) to generate maps, mosaics, orthophotos, point clouds etc. Several case studies have been published by http://drones.fsd.ch/en/ which should be considered to take into account.

AC – we thank the reviewer for this suggestion and we added these two paragraphs: From line 62 to line 70: Another important added value of RPAS is their adaptability that allows their use in various typologies of missions, and in particular for monitoring operations in remote and dangerous areas (Obanawa et al., 2014). The possibility to carry out flight operations at lower costs compared to ones required by traditional aircraft is also a fundamental advantage. Limited operating costs make these systems also convenient for multi-temporal applications where it is often necessary to acquire information on an active process (like a landslide) over the time. Beside their higher resolution and the possibility to extract reliable 3D information, UAV images are not conditioned by cloud cover as satellite imagery. A comparison between the use of satellite images, traditional aircraft and RPAS has been presented and discussed by Fiorucci et al. (2018) for landslides applications and by Giordan et al., (2018) for the identification of flooded areas. These contributions demonstrated the goodness of RPAS for on demand acquisitions of high resolution images over limited areas. from line 541 to line 544: "In particular during emergencies, the time required for RPAS dataset processing is an important element that should be carefully considered. Giordan et al. (2015a) presented a case study related to a landslide emergency. In this paper, authors considered not only possible results but also the time that is required for them."

RC -Line No. 45: "contest" or "context" of remote sensing research?

AC – context

RC - Line No. 48: SLR instead of RLS. I suggest to replace by "integrated camera systems" as well to address all kind of optical solutions for RPAS (i.e. bridge cameras, industrial grade cameras, video cameras, etc.).

AC – we modified the sentence: "In particular, the development of photogrammetry and technologies associated (i.e. integrated camera systems like compact cameras, industrial grade cameras, video cameras, single-lens reflex (SLR) digital cameras and GNSS/INS systems) allow to use of RPAS platforms in various applications as alternative to the traditional remote sensing method for topographic mapping or detailed 3D recording of ground information and a valid complementary solution to terrestrial acquisitions too (Nex and Remondino, 2014) (Fig.1)."

RC - Section from Line No. 52 to 62: I recommend to add the advantage of "micro RPAS are easy to transport into the disaster area". Foldable Systems (like DJI Mavic) fits easily into a day pack and can be transported safely as hand luggage. Weight matters especially for first responder teams like UNDAC or similar.

AC – line 55 to line 60: RPAS systems present some advantages in comparison to traditional platforms and, in particular, they could be competitive thanks to their versatility in the flight execution (Gomez and Purdie, 2016). Mini/micro RPAS are the most diffused for civil purposes, and they can fly at low altitudes according to limitations defined by national aviation security agencies and be easy transported into the disaster area. Foldable systems fits easily into a daypack and can be transported safely as hand luggage. This advantage is particularly important for first responder teams like UNDAC or similar.

RC - Section from line no. 83 to 104: I recommend to add some references to papers which analyses possible accuracies by using / not using GCPs and SFM Tools (i.e. Pix4D, Agisoft) or common photogrammetric workflows (i.e. Inpho Match AT). I suggest as well to add some references here to fast mosaicking methods - i.e. PhaseOne and IGI showed promising results with the commercial IGI Mapper System and the

German Aerospace Center developed specialized solutions for realtime traffic management (VABENE) and realtime mapping applications (MACS) on manned and unmanned aircrafts. Intro section in general: You name laser scanning and gas detection and also reference on that in section 2.1.1, 2.1.2, 2.1.3, 2.4.2 and 2.5 but a workflow description is missing. I recommend to add this workflow description or to specify the argumentation of using optical sensor systems.

AC – as we mentioned before, the principal aim of this manuscript is a review of available bibliography. We decided to avoid the publication of performance comparison between RPAS and/or software because we think that the focus is different. We mentioned papers like Remondino et al., (2014) and Nocerino et al., (2015) that considered this topic to complete our review. We thank for the suggestion about the rapid mapping and we added the following sentence: "In particular during emergencies, the time required for the image dataset processing can be a critical point. For this reason, the development of fast mosaicking methods as MACS, for a real time mapping applications, or VABENE++, developed by German Aerospace Center for real time traffic management (Detzer et al., 2015)."

RC - Line No. 128: Reference of (ADSR 2015) is missing. Update to ADSR 2016 is recommended.

AC – ADSR 2015 was already cited in bibliography as now ADSR 2016, with the suggested citation: Guha-Sapir, D., Hoyois, P., Wallemacq P. and Below, R.: Annual Disaster Statistical Review 2016 The numbers and trends. Centre for Research on the Epidemiology of Disasters, Ciaco Imprimerie, Louvain-la-Neuve (Belgium), pp. 91, 2017

RC - Section 2.1: I recommend to add the main parameters which influence the accuracy of derived DEM and orthophotos (i.e. real GSD, knowledge about interior and exterior orientation parameters, overlap of images, flight strip configuration and used SfM-Software)

AC – we added (line 119-121): In real applications, many parameters can influenced the final resolution of DSM/DTM and orthophoto like: real GSD (Nocerino et al., 2013) interior and exterior orientation parameters (Kraft et al., 2016), overlap of images, flight strip configuration and used SfM-MVS software (Nex et al., 2015).

RC - Line No. 281: First use of SfM-MVS - please explain.

AC – Structure from Motion-Multi View Stereo (SfM-MVS), we improved the text